# In-Domain Transfer Learning Strategy for Tumor Detection on Brain MRI

**DOI:** 10.3390/diagnostics13122110

**Published:** 2023-06-19

**Authors:** Duygu Sinanc Terzi, Nuh Azginoglu

**Affiliations:** 1Department of Computer Engineering, Amasya University, Amasya 05100, Turkey; duygu.terzi@amasya.edu.tr; 2Department of Computer Engineering, Kayseri University, Kayseri 38280, Turkey

**Keywords:** brain MRI, brain tumor detection, object detection, segmentation, transfer learning

## Abstract

Transfer learning has gained importance in areas where there is a labeled data shortage. However, it is still controversial as to what extent natural image datasets as pre-training sources contribute scientifically to success in different fields, such as medical imaging. In this study, the effect of transfer learning for medical object detection was quantitatively compared using natural and medical image datasets. Within the scope of this study, transfer learning strategies based on five different weight initialization methods were discussed. A natural image dataset MS COCO and brain tumor dataset BraTS 2020 were used as the transfer learning source, and Gazi Brains 2020 was used for the target. Mask R-CNN was adopted as a deep learning architecture for its capability to effectively handle both object detection and segmentation tasks. The experimental results show that transfer learning from the medical image dataset was found to be 10% more successful and showed 24% better convergence performance than the MS COCO pre-trained model, although it contains fewer data. While the effect of data augmentation on the natural image pre-trained model was 5%, the same domain pre-trained model was measured as 2%. According to the most widely used object detection metric, transfer learning strategies using MS COCO weights and random weights showed the same object detection performance as data augmentation. The performance of the most effective strategies identified in the Mask R-CNN model was also tested with YOLOv8. Results showed that even if the amount of data is less than the natural dataset, in-domain transfer learning is more efficient than cross-domain transfer learning. Moreover, this study demonstrates the first use of the Gazi Brains 2020 dataset, which was generated to address the lack of labeled and qualified brain MRI data in the medical field for in-domain transfer learning. Thus, knowledge transfer was carried out from the deep neural network, which was trained with brain tumor data and tested on a different brain tumor dataset.

## 1. Introduction

Medical image processing is an interdisciplinary field that bridges the domains of computer science, mathematics, and medicine. It involves the use of various modalities such as X-rays, computerized tomography (CT), magnetic resonance imaging (MRI), and positron emission tomography (PET) to generate images of different parts of the human body. Medical image processing encompasses the utilization of advanced algorithms and computational techniques to enhance and extract critical information from these images to aid in the diagnosis, treatment, and monitoring of various medical conditions [1].

Medical image processing has found significant application in the field of brain imaging, playing a crucial role in analyzing and interpreting brain-related medical images. The brain is a complex organ, and neuroimaging techniques have helped us better understand its structure, function, and disorders. The use of deep learning (DL) in brain imaging has enabled the identification of patterns and abnormalities that may go unnoticed by human observers, leading to earlier and more accurate diagnoses. Systems that previously relied on handcrafted features have evolved to systems that acquire features directly from data. DL has been employed in the preprocessing and analysis of brain images, encompassing various tasks, such as brain segmentation, disease diagnosis, lesion detection, registration, noise reduction, and resolution enhancement [2].

DL can continuously improve its accuracy as it learns from larger and more diverse datasets. This creates the need for large amounts of labeled medical data, which can be difficult and time consuming to obtain. Transfer learning (TL) reduces the required amount of labeled data and enhances model generalization. TL refers to the capability of a deep learning model to reuse knowledge learned from one task and apply it to a related but different task. Some of the benefits of TL include reduced training time and improved performance over models trained from scratch. The comparison of the traditional learning logic and TL is given in Figure 1.

Several studies have utilized TL to assist in the detection of brain tumors. When these studies are evaluated as seen in (Table 1), they generally focus on three tasks: classification, segmentation and object detection. The process of brain tumor classification involves categorizing tumors based on their characteristics and origins. Brain tumor segmentation aims to separate the tumor from the healthy brain tissue and detect the size, shape, location, and spread of the tumor. Brain object detection refers to the process of identifying different brain structures or objects, such as tumors, blood vessels, and nerves. The development of automated methods for tumor classification, segmentation, and object detection significantly reduces the workload of healthcare professionals and improves patient outcomes.

Asif et al. developed a method for multi-class brain tumor classification [3]. Bairagi et al. focused on tuning the hyperparameters and used TL for a faster training process [4]. Polat and Güngen classified brain tumors with TL networks using different optimization algorithms [5]. Hao et al. proposed a framework based on transfer learning and active learning to reduce the annotation cost with the uncertainty sampling method [6]. Ahmad and Choudhury analyzed transfer learning-based deep learning methods using traditional classifiers [7]. Ullah et al. presented a comparative analysis of several state-of-the-art pre-trained transfer learning techniques [8]. Deepak and Ameer utilized TL to extract features from brain MRI images for the purpose of classifying brain tumors [9]. Swaraja et al. proposed a framework that provides both brain tumor segmentation and classification, which can be efficiently processed on all four basic image pulse sequences [10]. Singh et al. developed a modified TL method to improve brain tumor segmentation accuracy [11]. Neubauer et al. presented a framework that transfers knowledge from human claustrum segmentation in adult images to neonatal scans [12]. Montalbo proposed TL techniques which consume fewer computational resources to detect three distinct brain tumors [13]. Shelatkar and Bansal compared different variants of YOLOv5 to detect brain tumor location [14]. Chegraoui et al. suggested a model that combines segmentation and detection tasks to find the locations of rare brain tumors [15].

Examining the source datasets employed in TL reveals that they are not necessarily directly relevant to the target task. For example, ImageNet and MS COCO have proven effective as source data in various tumor detection tasks. However, the use of natural images as a source data for describing medical images has become controversial recently [16]. Results seem to improve when the structure of the source images is similar to the target images. This situation is encountered not only in brain tumor detection but also in other medical tasks, such as the classification of parotid gland tumors [17], detection of hip fractures [18], and melanoma screening [19].

This study aims to make a quantitative comparison of transfer learning studies with in-domain transfer learning studies in the object detection task made with natural image datasets due to the scarcity of domain-specific datasets in the medical field. Thus, it is planned to prevent pre-acceptance and to support the in-domain thesis with numerical results.

To this end, in this study, the effect of both weight initialization and fine-tuning strategies is measured on both natural and medical datasets. In this context, models are trained with five different strategies, namely, random, MS COCO, BraTS, MS COCO + BraTS and MS COCO + BraTS + Fine Tune. The Gazi Brains 2020 data set is used as the target dataset. The success of the pre-trained models on the locating and segmenting of the tumor region as an object recognition task on brain MRIs is compared and reported. The most effective strategies determined using the Mask R-CNN backbone are also tested with YOLOv8, a state-of-the-art deep learning model. The obtained results show that even if the amount of data is less, TL between the same domains is more efficient than those that are cross domain in terms of both convergence and model performance. Data augmentation increases the performance, and it is required especially if cross-domain transfer learning is performed.

Section 2 details the datasets utilized in this study, their preparation, and the methodologies employed. Section 3 outlines the experimental setups and presents the experimental results. Lastly, Section 4 provides the conclusion and future works.

## 2. Materials and Methods

### 2.1. Transfer Learning

Transfer learning is actually modeling the concept of the human experience for use in the machine. Through the TL method, experience is transferred from successful models trained on large data sets. It can provide life-saving speed, especially in areas with data hunger and labeled data shortages.

The scientific definition of TL has been given in various publications [20,21] as a domain *D* consisting of two components as feature space *X* and a marginal probability distribution P(X), where $X=\{x_{1},\ldots ,x_{n}\}\in \chi$. Given a specific domain D={ꭓ,P(X)}, a task consists of two components, a label space γ and an objective predictive function f(•), denoted by T={γ,f(•)}, which can be learned from the training data pairs $\left\{x_{i},y_{i}\right\}$, where $x_{i}\in \chi$ and $y_{i}\in \gamma$. The f(•) can be used to predict the label of a new instance *x*, which can be rewritten by the probabilistic form of the conditional probability distribution P(Y|X). A task then can be defined as T={γ,P(Y|X)}. With the notations of domain and task, the TL is defined as follows.

Given a source domain $D_{S}$ and learning task $T_{S}$, and a target domain $D_{T}$ and learning task $T_{T}$, TL aims to help improve the learning of the target predictive function f(•) in $D_{T}$ using the knowledge in $D_{S}$ and $T_{S}$, where $D_{S}\neq D_{T}$ or $T_{S}\neq T_{T}$.

In the above definition, the $D_{S}\neq D_{T}$ condition means $X_{S}\neq X_{T}$ or $P_{S}\left(X\right)\neq P_{T}\left(X\right)$; that is, the source and target areas have different feature gaps or marginal probability distributions. Contrarily, the condition $T_{S}\neq T_{T}$ means either $Y_{S}\neq Y_{T}$ or $P\left(Y_{S}|X_{S}\right)\neq P\left(Y_{T}|X_{T}\right)$; that is, the source and target areas have different label spaces or conditional probability distributions. It should not be forgotten that with the target and source areas, i.e., $T_{S}=T_{T}$, the learning problem becomes a traditional ML problem. If TL improves the performance using only $D_{T}$ and $T_{T}$, the result is called positive transfer. Otherwise, TL corruption leads to a negative transfer.

There are two basic methods in TL: fine tuning and weight initializing [22]. In fine tuning, some layers of the model are frozen, training is performed with the new data, and the weights that have not been frozen are updated. Another method in fine tuning is to freeze all layers, except the output layer (this layer is known as classifier or fully connected layer). Fine tuning is also referred to as domain adaptation. In the weight initializing method, weights are taken directly from a pre-trained model, and then updated by re-training with new data. With these TL methods, experience in different tasks is transferred to the new problem (or task), and the training process is accelerated.

In this study, both the TL strategies weight initialization and fine-tuning mechanism were used as the learning transfer method since the TL effect was measured.

### 2.2. Datasets

In this section, the MS COCO, BraTS, and Gazi Brains datasets used during the experiments are introduced.

#### 2.2.1. MS COCO

Microsoft Common Objects in Context (MS COCO) [23] is a large-scale dataset that can be used for segmentation, captioning tasks, and especially object detection. This dataset contains 330,000 images in 80 categories, more than 200,000 of which are labeled. In this study, MS COCO was used as a source dataset in Strategy 2 and Strategy 4.

#### 2.2.2. BraTS 2020

Brain Tumor Image Segmentation Benchmark (BRATS) [24], which is defined as the MICCAI Multimodal Brain Tumor Segmentation Challenge 2020 dataset, contains MRI sequence data containing brain tumor patients (HGG and LGG). In this study, 293 HGG patient data included in BraTS 2020 are used for pre-training (source dataset) to measure the effect of TL from the same domain. The reason for using HGG data from the BraTS dataset is that the Gazi Brains 2020 dataset, which is another dataset we used, consists of only HGG data.

#### 2.2.3. Gazi Brains 2020

Gazi Brains 2020 is an MRI sequence dataset containing 50 normal and 50 histopathologically diagnosed HGG patients [25]. All patients have fluid-attenuated inversion-recovery (FLAIR), T1-weighted (T1w) and T2-weighted (T2w) sequences. All HGG patients and 12 normal patients also have gadolinium-enhanced T1w sequences. Gazi Brains 2020 has rich labeled data. Fifteen different labels consist of anatomical parts, such as region of interest (RoI), eye, optic nerve, lateral and third ventricle and tumor parts, such as peritumor edema, contrast enhancing— non-enhancing, necrosis. The data of 50 HGG patients included in this dataset were used for the target dataset to determine the effect of TL from the same domain.

Gazi Brains 2020 is a new brain MRI dataset generated to meet the needs for qualified MRI datasets. Although a few studies have started to use this dataset [26,27,28], it was used for the first time for the in-domain TL. In this study, slices from 50 patients with HGGs were used.

### 2.3. Data Preparing

In this study, the BraTS dataset was used for the TL source, and the Gazi Brains 2020 dataset was used for detecting the best weight initialization strategies (target dataset). Due to the characteristics of both datasets, the data-preparing pipeline varied. In Figure 2, the pipeline shown by the red arrow is used for Gazi Brains 2020, and the BraTS data preparing pipeline is shown in blue. Some examples corresponding to the relevant data preparing step are also given in Figure 2. All steps for data preparation are explained below.
MRI Sequence Selection: In this study, three different MRI sequences, such as FLAIR, T1w and T2w, were selected because they are default MRI sequences for both normal and HGG patients. Then, the input was prepared as three channels for each slice.Registration: Different sequences of an MRI can be in different orientations and sizes. Therefore, the FLAIR sequence was used as a template, and other sequences (T1w and T2w) were spatially aligned according to the template sequence. Advanced normalization tools (ANTs) software package [29] with the rigid transformation method were used for the registration process.RoI Extraction: Skull stripping was performed for each slice so that the models were focused only on the brain tissue. Thus, bias originating from the skull was prevented for models. The segmentation information provided with the Gazi Brains 2020 data set was used to perform skull stripping. From the segmentation data, the binary RoI masks were extracted using Label-0 (clear label), Label-2 (eye) and Label-3 (optic nerve), and RoI information was extracted from the images using these masks.Generating bounding boxes and masks: Instance-based bounding box and mask production were carried out due to the use of the Mask R-CNN model within the scope of this study. Both whole tumor (peritumor edema, contrast enhancing—non enhancing, necrosis) and brain tissue were given as objects to the Mask R-CNN model. Brain tissue was also considered an object in this study since brain tissue looks very different in the starting and ending slices due to the nature of the brain MRI. Regionprops [30] was used for the production of instance-based bounding boxes and masks. As shown in Figure 2, the bounding box belonging to each component of the tumor and brain tissue and the mask information belonging to these bounding boxes were extracted and given for training the models.Padding: Before resizing, the padding process was carried out to preserve the symmetry of the MRI.Data Augmentation: The data augmentation process with probability *p* = 0.5 was performed during the model training for the training pipeline with augmentation. Thus, more generalizable models were developed in the relatively small Gazi Brains 2020 dataset.Resizing: Images were resized to 240 × 240 since the BraTS dataset has MRI in the specified sizes.

### 2.4. Deep Learning Architecture

Two basic models were used within the scope of the study. The first of these is Mask R-CNN. Mask R-CNN, a state-of-the-art object detection and segmentation model, was used in this study. This model outputs the class labels, the bounding box of the object, and the object mask. This model uses ResNet-101 as ConvNet for feature extraction and also uses the regional proposal network (RPN) for RoI generation while detecting the candidate bounding boxes. The Mask R-CNN method is the extended version of the Faster R-CNN [31] and is given in Figure 3.

The best-performing strategies determined in Mask R-CNN were also tested in the YOLOv8 [32] model. The Mask R-CNN results showed that data augmentation gave better results in any case. For this reason, YOLOv8 experiments were performed using data augmentation. In addition, the YOLOv8 model, which does not have a fine-tuning mechanism in its implementation, was used without changing the original form.

### 2.5. Transfer Learning Methodology

Within the scope of this study, five different weight initialization strategies were tested to determine the best strategy on the medical object detection task (Figure 4). These strategies are as follows.
Strategy 1: Initializing Mask R-CNN model with random weights on Gazi Brains 2020 dataset. The pre-trained model is not used in this strategy.Strategy 2: Initializing Mask R-CNN model with MS COCO weights. Accordingly, the initial (pre-trained) weights are obtained from the model trained on the MS COCO dataset. Only the output of the classifier is updated as two classes as tumor and brain.Strategy 3: Initializing Mask R-CNN model with the weights trained on the BraTS dataset using random initial weights. In this strategy, the weights of the model trained on the BraTS dataset are used as starting weights.Strategy 4: Initializing Mask R-CNN model with the weights trained on the BraTS dataset using MS COCO initial weights. In this strategy, different from Strategy 3, resource TL model is used on the BraTS dataset using MS COCO initial weights.Strategy 5: Unlike Strategy 4, all layers except the classifier are frozen, and the classifier part is fine tuned.

For the default and augmentation, two different pipelines were used to measure the effects of the initial weight strategies on the Mask R-CNN model. In the first one, all training hyperparameters were kept constant for all strategies, then models were trained with these parameters. In the second, data augmentation was performed for all strategies, and the ideal learning rate was found using the learning rate finder. Then, models were trained using the augmentation pipeline. Data augmentation is used to increase the generalization capacity of models on relatively small datasets. Thus, more successful models are developed by regulating models with data.

The most effective strategies, which were determined by evaluating the results obtained using the Mask R-CNN model, were also tested using another object detection model, YOLOv8. YOLOv8 experiments were performed by applying data augmentation to the data considering Mask R-CNN experiments, and fine tuning was not performed based on the original model.

All hyperparameters for all strategies in the default pipeline are the same. For the models used in the default pipeline, the hyperparameters used included epoch number 100, batch size 8, optimizer stochastic gradient descent, and learning rate 5 × 10${}^{-3}$. The models were also trained by dividing the learning rate in half for each 25 epoch using the learning rate scheduler. For the augmented pipeline, all hyperparameter settings for all strategies were the same with the default parameters, except the learning rate. Since datasets change in each fold of each strategy, the learning rate specific to each fold was found using the learning rate finder [33]. All learning rates for each fold are given in Table 2. In the augmented pipeline, a dynamic learning rate scheduler was used to adjust the learning rate while training. Some data augmentation techniques, such as vertical flip, random brightness (0.15), contrasts (0.15), shift (0.0625), scale (0.2) and rotate (15), were used with 0.5 probability. Values in parentheses indicate the related augmentation limit.

In particular, models that start training with random weights may have various disadvantages, such as later convergence, sticking to local minima, etc., compared to models trained with weights taken from pre-trained models. Data augmentation and the ideal learning rate were found (Table 2) in this study to reduce the effect of these disadvantages on the model training process.

## 3. Results

All experiments were implemented in PyTorch 1.12.1. The following libraries were used in the analysis: Python scikit-learn for data splitting and Python seaborn for data visualization.

To measure the effect of five different weight initialization strategies determined within the scope of this study, the 5-fold cross-validation method was used, and the results were visualized. In Figure 5, models were trained without data augmentation. The upper part for the bounding box results and the lower part for the segmentation results are given in Figure 5 for the tumor class. The *X*-axis represents the epoch, and the *Y*-axis represents the intersection over union (IoU) value corresponding to the specified metric. In Figure 6, different from Figure 5, data augmentation is performed during training, and learning rate optimization is also performed using the learning rate finder. In Figure 5 and Figure 6, the AP@[0.5:0.95] metric for tumor class is used as the success metric for both the bounding box and segmentation results. The AP@[0.5:0.95] metric expresses the AP (average precision) for IoU thresholds from 0.5 to 0.95 by the step size of 0.05.

Mean values of the 5-fold cross-validation method according to three different metrics for both the bounding box and segmentation results are presented in Figure 7 for the tumor class. The metrics used in Figure 7 are given as AP@[0.5:0.95], AP@[0.5] and AR@[0.5:0.95], respectively. In the figure, the *X*-axis represents the metric used, and the *Y*-axis represents the IoU value. The augmentation effect is also visualized as an overlay plot with a dashed and transparent format.

According to the mean values of the 5-fold cross-validation method, as seen in Figure 5 and Figure 6, Strategy 4 gives better bounding boxes and segmentation IoU values than other strategies. The bounding boxes and segmentation results are given below. Values in parentheses indicate the augmented pipeline training results. All cross-validation results for the bounding box are given in Table 3, and the segmentation is shown in Table 4.
As given in Table 5, when the warm-up processes of the models, which are the averages of the first five epochs of five folds, were examined, Strategy 4 has 7% (10%), 24% (41%) and 40% (42%) higher AP@[0.5:0.95] than Strategies 1–3, respectively, and 6% (19%) is higher than Strategy 5. Additionally, Table 5 shows that extensive augmentation is not needed for the fine-tuning strategy.Bounding Box Results: For the AP@[0.5:0.95] metric, Strategy 4 yielded 4% (3%), 10% (7%), 17% (10%), and 2% (5%) higher IoU values, respectively, than the other four strategies (Strategies 3, 2, 1 and 5). Similarly, for the AP@[0.5] metric, Strategy 4 yielded 2% (1%), 11% (7%), 12% (7%), and 2% (3%) higher IoU values, respectively than the other four strategies. Additionally, for the AR@[0.5:0.95] metric, Strategy 4 yielded 5% (3%), 9% (4%), 17% (10%), and 3% (4%) higher IoU values than the other four strategies, respectively.Segmentation Results: For the AP@[0.5:0.95] metric, Strategy 4 yielded 3% (3%), 1% (8%), 15% (1%), and 3% (4%) higher IoU values, respectively, than the other four strategies (Strategy 3, 2, 1, and 5). Similarly, for the AP@[0.5] metric, Strategy 4 yielded 2% (1%), 11% (5%), 14% (6%), and 2% (3%) higher IoU values, respectively, than the other three strategies. Additionally, for the AR@[0.5:0.95] metric, Strategy 4 yielded 3% (2%), 6% (3%), 14% (8%), and 2% (4%) higher IoU values than the other four strategies, respectively.When the strategies are examined according to the generally accepted value of IoU 0.5 in the literature, there is no difference between Strategy 1 and Strategy 2, as they are 73% vs. 73% AP@[0.5] for bounding boxes in the augmented pipeline. Additionally, the AP@[0.5] metric values of the segmentation are very close to each other as 73% vs. 74%. At the same time, the AP@[0.5] results for bounding boxes obtained from the default pipeline are also close to each other as 67% vs. 68%.Although there is no difference between Strategy 1 and Strategy 2 according to the frequently used AP@[0.5] metric, Strategy 2 has 7% (3%) higher IoU value in terms of the AP@[0.5:0.95] metric for bounding boxes.When the effect of data augmentation on the models is measured, data augmentation is more effective in Strategy 1 and Strategy 2 compared to other strategies. Thus, considering the metric AP@[0.5:0.95], there is a 9% increase in the value of AP for the bounding box in Strategy 1 and a 5% increase in Strategy 2, while this ratio is 3%, 2%, and 1% for Strategies 3, 4, and 5, respectively.When Strategies 3 and 4 are examined, the source weights in both strategies come from the same domain (brain MRI). Strategy 4, unlike Strategy 3, uses pretrained MS COCO weights during training, and its success rates are better than Strategy 3 for all 3 metrics. For example, in default pipeline, there is a 4% difference between the bounding box AP@[0.5:0.95] of Strategy 4 and Strategy 3. However, considering the AP@[0.5] metric, there is only 2% (1%) difference between these two strategies.The difference between Strategies 4 and 5 is that the classification layer of Strategy 5 is frozen and fine tuned. Although the results of Strategy 4 and Strategy 5 are close to each other, it is seen that Strategy 4 achieves better results: 2% (5%) for the bounding box AP@[0.5:0.95] metric, 2% (3%) for the bounding box AP@[0.5] metric, and 3% (4%) for the bounding box AP@[0.5:0.95] metric, respectively.

In addition, the proposed strategies were also tested using YOLOv8. YOLOv8 was used in Strategy 1 (random initialization), Strategy 2 (MS COCO), and Strategy 4 (MS COCO + BraTS), as its original implementation does not have a fine-tuning mechanism. Mean values of the 5-fold cross-validation method for both the bounding box and segmentation results are presented in Figure 8. The augmented pipeline training results are given below. As with the results obtained using Mask R-CNN, the best results with YOLOv8 were achieved with Strategy 4. All cross-validation results for the bounding box are given in Table 6, and the segmentation is shown in Table 7.
Bounding Box Results: For the AP>0.5 metric, Strategy 4 yielded 6% and 10% higher IoU values than Strategy 2 and Strategy 1, respectively.Segmentation Results: For the AP>0.5 metric, Strategy 4 yielded 7% and 10% higher IoU values than Strategy 2 and Strategy 1, respectively.

## 4. Conclusions and Future Works

Although the effect of TL in the medical field is measured in various classical detection models, studies measuring the effect of TL for the object detection task are not available in the literature according to the best of our knowledge. We scientifically measured the performance of TL strategies for medical object detection.

In this study, five different scenarios involving the use of weights obtained from random and various include domain (medical image dataset) and cross-domain (natural image dataset) datasets were created. It was observed that classical MS COCO-based pre-trained models do not have much effect on the success rates according to frequently used metrics in the literature. In addition, data augmentation is a good alternative for the medical field that suffers from data shortage. We also confirmed that medical object recognition models converged more quickly, thanks to TL from a similar domain as stated in Table 5.

Two different data processing pipelines were also used in this study. As seen in Figure 7, data augmentation techniques were more effective in models trained with TL from the cross domain as expected. This is due to the very limited ability of the MS COCO dataset to represent the medical field. Thanks to the TL from the medical field, the object definition is limited to the medical field, and it does not require much data augmentation to increase the generalization capacity of the model compared to the classical MS COCO and random weight initialization strategies.

Although there is no difference between the MS COCO and random weights initialization strategy according to the AP@[0.5] metric frequently used in the object detection literature, the results for these two strategies differ according to another metric, AP@[0.5:0.95]. Since AP@[0.5:0.95] and AR@[0.5:0.95] metrics give the average of the results for many IoU thresholds, they give more sensitive results for tumor object detection for all range of IoU values. According to these results, the MS COCO weights initialization strategy outperforms the random weights initialization strategy in terms of finding smaller tumor objects, detecting tumor objects at higher IoU thresholds and reducing the false negative. In addition, the proposed strategies were also implemented with YOLOv8, and similar trends were obtained.

There are a few limitations in this study due to the model and data set:Only the BraTS dataset was used as the targeted TL source. In the BraTS dataset, only HGG data were used to be compatible with the Gazi Brains 2020 dataset.Both weight initialization and fine-tuning strategies were studied as the TL mechanism.Since this study aims to determine different initial weight strategies, only the Mask-RCNN and YOLOv8 models were used to compare the initial weight strategies.Only the hyperparameters detailed in Section 2.5 were used in terms of time and computational cost.

Future works will aim to minimize or eliminate any existing limitations as much as possible. Moreover, it is planned to measure the effect of TL in the health domain but using a different data type (different image modality and/or different medical data).

## Figures and Tables

**Figure 1 diagnostics-13-02110-f001:**
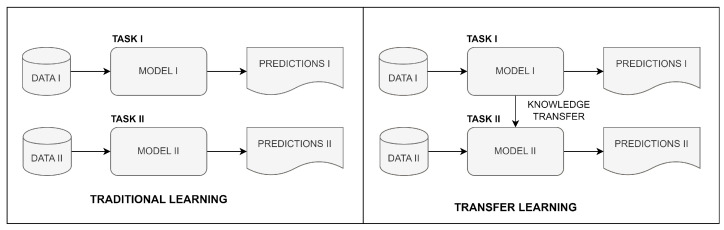
Comparison of traditional and transfer learning.

**Figure 2 diagnostics-13-02110-f002:**
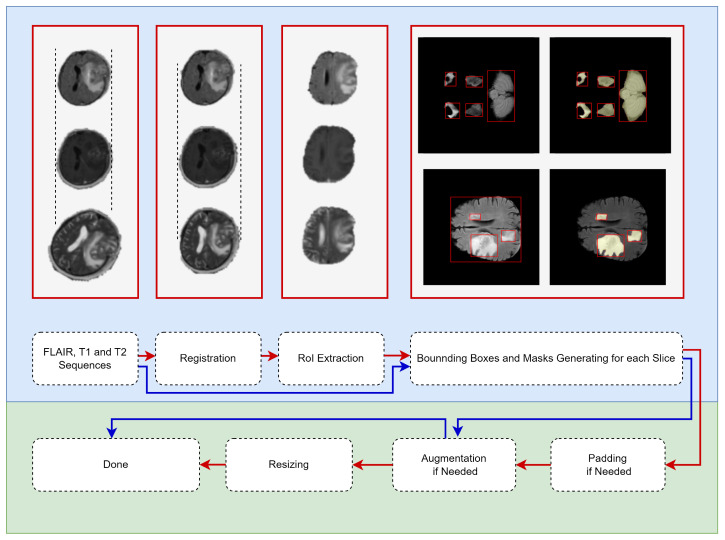
Data-prepa ring pipeline for medical object detection. (Red flow for Gazi Brains 2020 Dataset and, Blue flow for BraTS Dataset. The red boxes are put as examples of the operations written under them.)

**Figure 3 diagnostics-13-02110-f003:**
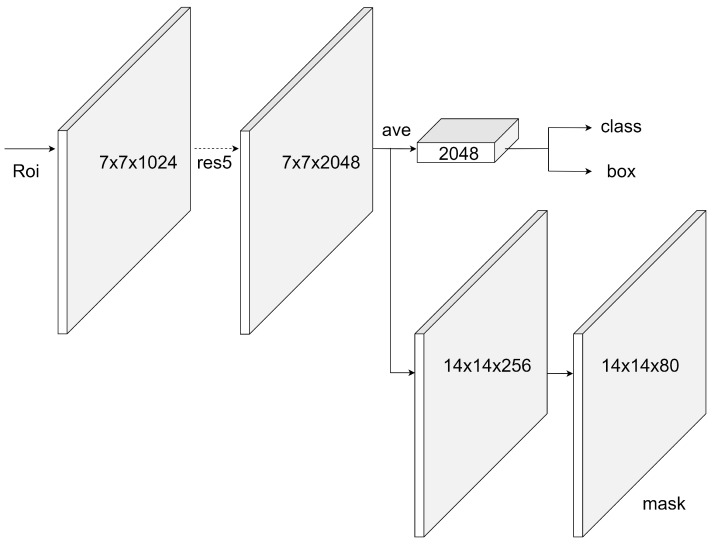
Mask R-CNN with ResNet backbone.

**Figure 4 diagnostics-13-02110-f004:**
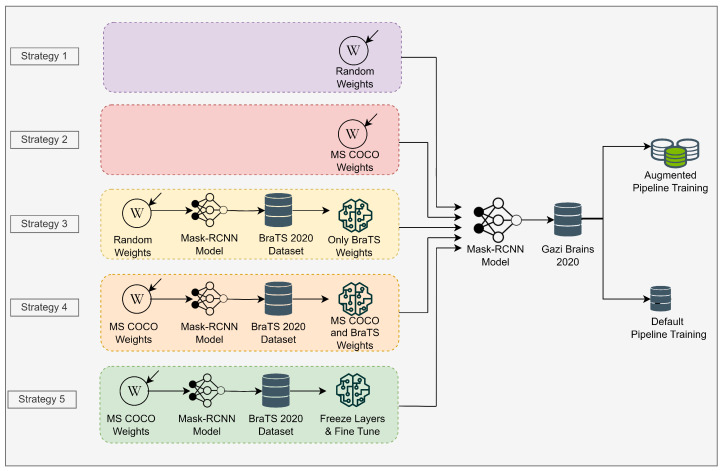
Transfer learning strategies in the scope of this study.

**Figure 5 diagnostics-13-02110-f005:**
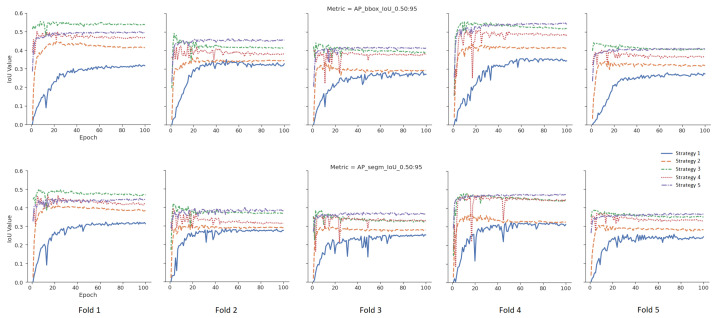
Validation results of the default pipeline for each strategy.

**Figure 6 diagnostics-13-02110-f006:**
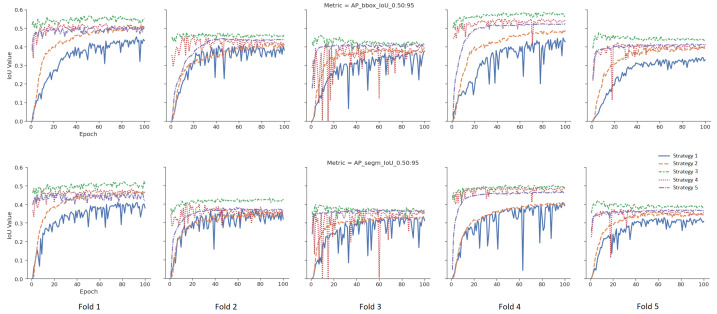
Validation results of the augmented pipeline for each strategy.

**Figure 7 diagnostics-13-02110-f007:**
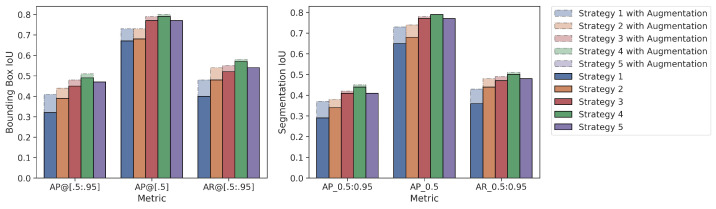
Average values of 5-fold cross validation results for Mask R-CNN.

**Figure 8 diagnostics-13-02110-f008:**
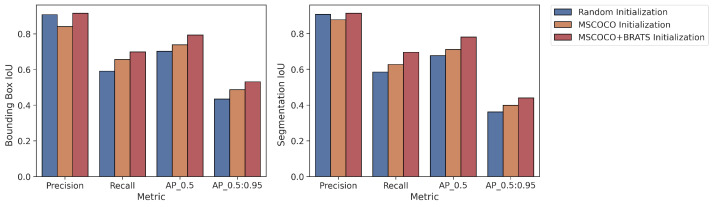
Average values of 5-fold cross validation results for YOLOv8.

**Table 1 diagnostics-13-02110-t001:** Comparative analysis for transfer learning studies on brain tumor detection.

Reference	Task	TL Strategy	DL Architecture	Best Result	Source Dataset	Target Dataset
[3]	C	FT	Xception + DDB	Accuracy of 99.67%	ImageNet	Figshare Dataset, Kaggle Dataset
[4]	C	FT	AlexNet	Accuracy of 98.67%	ImageNet	BRATS 2013, BRATS 2015, OPEN I
[5]	C	FT	ResNet50	Accuracy of 99.02%	ImageNet	Figshare Dataset
[6]	C	FT	AlexNet	AUC of 82.89%	ImageNet	BRATS 2019
[7]	C	FT	VGG-19-SVM	Accuracy of 99.39%	ImageNet	Kaggle Dataset
[8]	C	FT	Inception ResNetV2	Accuracy of 98.91%	ImageNet	Kaggle Dataset
[9]	C	FT	GoogLeNet + KNN	Accuracy of 98.0 ± 0.4%	ImageNet	Figshare Dataset
[10]	C, S	FT	GoogLeNet	*	ImageNet	BRATS 2013, BRATS 2015, BRATS 2017, Figshare Dataset
[11]	S	WI	3D U-Net	Mean IOU of 0.8803 ± 0.020	BraTS 2018, BraTS 2019, BraTS 2020	BraTS 2021
[12]	S	WI	U-Net	Dice score 80.0%	dHCP (adult)	dHCP (neonatal)
[13]	OD	FT	YOLOv4-Tiny	mAP of 93.14	MS COCO	Figshare Dataset
[14]	OD	FT	YOLOv5x	mAP of 91.2%	MS COCO	BraTS 2021
[15]	OD, S	FT	YOLO	*	MS COCO	BraS 2019, DIPG
Proposed Approach	OD, S	WI, FT	Mask R-CNN, YOLOv8	**		

C: classification; S: segmentation; OD: object detection; FT: fine tuning; WI: weight initialization. *: For detailed comparison of results, check the relevant paper. **: Presented in the Results Section.

**Table 2 diagnostics-13-02110-t002:** Learning rates used for each fold and strategy.

Fold	Strategy 1	Strategy 2	Strategy 3	Strategy 4	Strategy 5
1	6.31 × 10${}^{-3}$	1.50 × 10${}^{-3}$	3.11 × 10${}^{-3}$	1.07 × 10${}^{-2}$	1.07 × 10${}^{-2}$
2	6.61 × 10${}^{-3}$	1.10 × 10${}^{-3}$	6.11 × 10${}^{-3}$	4.01 × 10${}^{-4}$	4.01 × 10${}^{-4}$
3	5.61 × 10${}^{-3}$	1.30 × 10${}^{-3}$	1.15 × 10${}^{-2}$	3.00 × 10${}^{-3}$	3.00 × 10${}^{-3}$
4	5.51 × 10${}^{-3}$	9.02 × 10${}^{-4}$	9.02 × 10${}^{-4}$	7.02 × 10${}^{-4}$	7.02 × 10${}^{-4}$
5	5.91 × 10${}^{-3}$	1.30 × 10${}^{-3}$	9.31 × 10${}^{-3}$	4.61 × 10${}^{-3}$	4.61 × 10${}^{-3}$

**Table 3 diagnostics-13-02110-t003:** Bounding box results for Mask R-CNN.

	Strategy 1			Strategy 2			Strategy 3			Strategy 4			Strategy 5		
**Folds**	** *AP* ** **@[0.5:0.95]**	** AP>0.5 **	* **AP** * **@[0.5:0.95]**	* **AP** * **@[0.5:0.95]**	** AP>0.5 **	* **AP** * **@[0.5:0.95]**	* **AP** * **@[0.5:0.95]**	** AP>0.5 **	* **AP** * **@[0.5:0.95]**	* **AP** * **@[0.5:0.95]**	** AP>0.5 **	* **AP** * **@[0.5:0.95]**	* **AP** * **@[0.5:0.95]**	** AP>0.5 **	* **AP** * **@[0.5:0.95]**
Folds 1	0.32 (0.45)	0.69 (0.79)	0.42 (0.52)	0.45 (0.51)	0.79 (0.81)	0.52 (0.59)	0.50 (0.53)	0.83 (0.85)	0.58 (0.60)	0.55 (0.57)	0.86 (0.87)	0.61 (0.62)	0.50 (0.51)	0.85 (0.86)	0.57 (0.58)
Fold 2	0.35 (0.42)	0.75 (0.80)	0.42 (0.51)	0.36 (0.42)	0.65 (0.74)	0.45 (0.55)	0.43 (0.46)	0.84 (0.84)	0.52 (0.55)	0.49 (0.48)	0.83 (0.83)	0.59 (0.56)	0.46 (0.44)	0.80 (0.77)	0.53 (0.52)
Fold 3	0.29 (0.38)	0.62 (0.67)	0.36 (0.43)	0.34 (0.39)	0.60 (0.68)	0.46 (0.47)	0.40 (0.42)	0.69 (0.72)	0.46 (0.48)	0.44 (0.46)	0.71 (0.72)	0.49 (0.52)	0.42 (0.42)	0.70 (0.70)	0.49 (0.49)
Fold 4	0.36 (0.45)	0.69 (0.75)	0.43 (0.52)	0.43 (0.49)	0.73 (0.76)	0.54 (0.60)	0.51 (0.55)	0.81 (0.82)	0.58 (0.61)	0.55 (0.58)	0.83 (0.84)	0.62 (0.65)	0.55 (0.52)	0.82 (0.80)	0.62 (0.60)
Folds 5	0.28 (0.34)	0.59 (0.66)	0.35 (0.42)	0.35 (0.40)	0.62 (0.68)	0.44 (0.50)	0.40 (0.42)	0.69 (0.73)	0.48 (0.50)	0.44 (0.47)	0.72 (0.74)	0.52 (0.55)	0.41 (0.42)	0.70 (0.70)	0.48 (0.49)
Mean	0.32 (0.41)	0.67 (0.73)	0.40 (0.48)	0.39 (0.44)	0.68 (0.73)	0.48 (0.54)	0.45 (0.48)	0.77 (0.79)	0.52 (0.55)	**0.49 (0.51)**	**0.79 (0.80)**	**0.57 (0.58)**	0.47 (0.46)	0.77 (0.77)	0.54 (0.54)

**Table 4 diagnostics-13-02110-t004:** Segmentation results for Mask R-CNN.

	Strategy 1			Strategy 2			Strategy 3			Strategy 4			Strategy 5		
**Folds**	* **AP** * **@[0.5:0.95]**	** AP>0.5 **	* **AP** * **@[0.5:0.95]**	* **AP** * **@[0.5:0.95]**	** AP>0.5 **	* **AP** * **@[0.5:0.95]**	* **AP** * **@[0.5:0.95]**	** AP>0.5 **	* **AP** * **@[0.5:0.95]**	* **AP** * **@[0.5:0.95]**	** AP>0.5 **	* **AP** * **@[0.5:0.95]**	* **AP** * **@[0.5:0.95]**	** AP>0.5 **	* **AP** * **@[0.5:0.95]**
Folds 1	0.32 (0.41)	0.69 (0.81)	0.40 (0.46)	0.41 (0.47)	0.78 (0.84)	0.49 (0.55)	0.47 (0.48)	0.85 (0.88)	0.51 (0.54)	0.50 (0.52)	0.88 (0.88)	0.55 (0.57)	0.45 (0.46)	0.85 (0.87)	0.51 (0.52)
Fold 2	0.29 (0.37)	0.70 (0.78)	0.35 (0.44)	0.32 (0.36)	0.67 (0.76)	0.42 (0.46)	0.40 (0.41)	0.79 (0.82)	0.47 (0.49)	0.42 (0.43)	0.82 (0.80)	0.50 (0.48)	0.40 (0.38)	0.76 (0.72)	0.46 (0.43)
Fold 3	0.26 (0.33)	0.60 (0.65)	0.33 (0.38)	0.30 (0.33)	0.61 (0.65)	0.41 (0.43)	0.36 (0.37)	0.69 (0.70)	0.41 (0.43)	0.39 (0.40)	0.69 (0.71)	0.44 (0.46)	0.37 (0.37)	0.70 (0.70)	0.44 (0.43)
Fold 4	0.33 (0.41)	0.69 (0.77)	0.39 (0.47)	0.37 (0.41)	0.73 (0.76)	0.48 (0.52)	0.47 (0.49)	0.81 (0.82)	0.53 (0.55)	0.48 (0.50)	0.83 (0.84)	0.54 (0.57)	0.48 (0.47)	0.82 (0.80)	0.55 (0.54)
Folds 5	0.26 (0.33)	0.57 (0.66)	0.32(0.40)	0.31 (0.35)	0.61 (0.68)	0.39 (0.45)	0.37 (0.37)	0.71 (0.69)	0.44 (0.44)	0.39 (0.42)	0.72 (0.74)	0.45 (0.48)	0.37 (0.37)	0.70 (0.70)	0.42 (0.43)
Mean	0.29 (0.37)	0.65 (0.73)	0.36 (0.43)	0.34 (0.38)	0.68 (0.74)	0.44 (0.48)	0.41 (0.42)	0.77 (0.78)	0.47 (0.49)	**0.44 (0.45)**	**0.79 (0.79)**	**0.50 (0.51)**	0.41 (0.41)	0.77 (0.76)	0.48 (0.47)

**Table 5 diagnostics-13-02110-t005:** Average AP@[0.5:0.95] results of first five epochs for bounding box and segmentation.

	*BBOX AP*@[0.5:0.95]		*SEGM AP*@[0.5:0.95]	
	**No Aug. Pip.**	**Aug. Pip.**	**No Aug. Pip.**	**Aug. Pip.**
Strategy 1	0.04	0.05	0.04	0.05
Strategy 2	0.20	0.06	0.21	0.06
Strategy 3	0.37	0.37	0.33	0.34
Strategy 4	0.44	0.47	0.38	0.41
Strategy 5	0.38	0.28	0.36	0.29

**Table 6 diagnostics-13-02110-t006:** Bounding box results for YOLOv8.

	Strategy 1				Strategy 2				Strategy 4			
**Folds**	**Precision**	**Recall**	** AP>0.5 **	* **AR** * **@[0.5:0.95]**	**Precision**	**Recall**	** AP>0.5 **	* **AR** * **@[0.5:0.95]**	**Precision**	**Recall**	** AP>0.5 **	* **AR** * **@[0.5:0.95]**
Folds 1	0.87	0.70	0.77	0.49	0.90	0.77	0.83	0.55	0.94	0.79	0.85	0.58
Fold 2	0.95	0.56	0.73	0.40	0.86	0.65	0.77	0.44	0.89	0.72	0.82	0.50
Fold 3	0.90	0.55	0.66	0.40	0.84	0.57	0.67	0.44	0.92	0.62	0.72	0.46
Fold 4	0.97	0.64	0.75	0.51	0.88	0.72	0.80	0.60	0.92	0.77	0.86	0.63
Folds 5	0.86	0.50	0.61	0.38	0.72	0.56	0.63	0.40	0.91	0.59	0.72	0.49
Mean	0.91	0.59	0.70	0.44	0.84	0.66	0.74	0.49	**0.92**	**0.70**	**0.80**	**0.54**
std	0.05	0.08	0.07	0.06	0.07	0.09	0.09	0.08	0.02	0.09	0.07	0.07

**Table 7 diagnostics-13-02110-t007:** Segmentation results for YOLOv8.

	Strategy 1				Strategy 2				Strategy 4			
**Folds**	**Precision**	**Recall**	** AP>0.5 **	* **AR** * **@[0.5:0.95]**	**Precision**	**Recall**	** AP>0.5 **	* **AR** * **@[0.5:0.95]**	**Precision**	**Recall**	** AP>0.5 **	* **AR** * **@[0.5:0.95]**
Folds 1	0.83	0.67	0.74	0.40	0.92	0.78	0.83	0.45	0.95	0.80	0.86	0.48
Fold 2	0.95	0.54	0.66	0.32	0.82	0.56	0.65	0.35	0.90	0.70	0.79	0.42
Fold 3	0.92	0.56	0.65	0.34	0.80	0.61	0.67	0.35	0.89	0.62	0.68	0.37
Fold 4	0.97	0.64	0.74	0.42	0.88	0.72	0.78	0.49	0.91	0.76	0.85	0.53
Folds 5	0.88	0.50	0.59	0.33	0.97	0.46	0.62	0.36	0.91	0.60	0.73	0.40
Mean	0.91	0.580	0.68	0.36	0.88	0.63	0.71	0.40	**0.91**	**0.70**	**0.78**	**0.44**
std	0.06	0.07	0.06	0.04	0.07	0.13	0.09	0.07	0.02	0.09	0.07	0.07

## Data Availability

BraTS 2020 and Gazi Brains 2020 datasets were used in the study. BraTS is the cult dataset used in this field [24]. Gazi Brains 2020 [25] (https://doi.org/10.7303/syn22159468, accessed on 1 June 2023) was produced especially against the labeled data problem in brain MRI and medical imaging and is a dataset obtained from original hospital MR images.

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
