# Peer review of "In-Domain Transfer Learning Strategy for Tumor Detection on Brain MRI"

_diagnostics, 2023, doi:10.3390/diagnostics13122110_

Round 1

Reviewer 1 Report

In this study, the 3 effect of transfer learning for medical object detection was quantitatively compared using natural 4 and medical image datasets. Within the scope of this study, transfer learning strategies based on 5 five different weight initialization methods were discussed. A natural image dataset MS COCO and 6 brain tumor dataset BraTS 2020 were used as the transfer learning source, and Gazi Brains 2020 was 7 used for the target. Mask R-CNN, a state-of-the-art object detection model, is used as deep learning 8 architecture. Experimental results show that transfer learning from the medical image dataset was 9 found to be 10% more successful and showed 24% better convergence performance than the MS 10 COCO pre-trained model although it contains fewer data. While the effect of data augmentation on 11 the natural image pre-trained model was 5%, the same domain pre-trained model was measured as 12 2%. According to the most widely used object detection metric, transfer learning strategies using 13 MS COCO weights and random weights showed the same object detection performance with data 14 augmentation. Results show that even if the amount of data is less than the natural dataset, in-domain 15 transfer learning is more efficient than cross-domain transfer learning. This is very interesting study the authors performed.  I have the following suggestion to improve :

1. Please specify the importance of the proposed solution.

2. The listed contributions are a little bit weak. Please highlight the novelty of the proposed solution.

3. What is the main question addressed by the research?  What does it add to the subject area compared with other published material?

4.  Although the domain transfer is good concept but how does the concept came and the usefullness / or limitation of traditional domain knowledge based methods differ need to explain. Also suggesting you add this paper in this introduction.
5. Please discuss how to obtain the suitable parameter values used in the solution. The experimental results are great. Please compare with more recently published solutions as backbone, especially the solutions published in 2023 and 2022. or add in the LR:  1) Pal, Debojyoti, Pailla Balakrishna Reddy, and Sudipta Roy. "Attention UW-Net: A fully connected model for automatic segmentation and annotation of chest X-ray." Computers in Biology and Medicine 150 (2022): 106083.  2) Roy, Sudipta, Tanushree Meena, and Se-Jung Lim. "Demystifying supervised learning in healthcare 4.0: A new reality of transforming diagnostic medicine." Diagnostics 12, no. 10 (2022): 2549.

6. Figure 5 and Figure 6 are clearly visible. Kindly use the some standard picture editing tools to enhance the quality of the paper. 

7. How does domain transfer will help for new data/ uncommon disease ?

Author Response

Thank you very much for your time and effort to review our article.

Comment 1:  Please specify the importance of the proposed solution.

Response 1: Additions have been made to the Introduction section to emphasize the importance of the proposed solution. In summary, instead of transfer learning (pre-training) with natural datasets consisting of thousands of images, transfer learning with few and domain-specific data gave better results. Thus, better results can be obtained by using less field-specific data and data augmentation together.

Comment 2: The listed contributions are a little bit weak. Please highlight the novelty of the proposed solution.

Response 2: Additions have been made to the “Introduction” and “Conclusion and Future Works” Sections.

Comment 3:  What is the main question addressed by the research?  What does it add to the subject area compared with other published material?

Response 3: The literature review has been updated. A table for comparison with the related studies has been added.

Comment 4: Although the domain transfer is good concept but how does the concept came and the usefullness / or limitation of traditional domain knowledge based methods differ need to explain. Also suggesting you add this paper in this introduction.

Response 4: The literature review has been updated. A table for comparison with the related studies has been added.

Comment 5:  Please discuss how to obtain the suitable parameter values used in the solution. The experimental results are great. Please compare with more recently published solutions as backbone, especially the solutions published in 2023 and 2022. or add in the LR:  1) Pal, Debojyoti, Pailla Balakrishna Reddy, and Sudipta Roy. "Attention UW-Net: A fully connected model for automatic segmentation and annotation of chest X-ray." Computers in Biology and Medicine 150 (2022): 106083.  2) Roy, Sudipta, Tanushree Meena, and Se-Jung Lim. "Demystifying supervised learning in healthcare 4.0: A new reality of transforming diagnostic medicine." Diagnostics 12, no. 10 (2022): 2549.

Response 5: These valuable studies that you have suggested have been cited and also new experiments results obtained by Yolov8, a state-of-the-art model, were added and included in the paper.

Comment 6: Figure 5 and Figure 6 are clearly visible. Kindly use the some standard picture editing tools to enhance the quality of the paper. 

Response 6: All figures have been reviewed and set to a minimum dpi of 300.

Comment 7: How does domain transfer will help for new data/ uncommon disease ?

Response 7: We know that there is a data set problem in such studies (new data/ uncommon disease). We realized in our review study that it would be beneficial to use different medical datasets instead of natural image datasets when data access is limited [1]. For example, for a tumor disease for which data cannot be found, different tumor data gives more successful results than natural image datasets.

Atasever, S., Azginoglu, N., Terzi, D. S., & Terzi, R. (2022). A comprehensive survey of deep learning research on medical image analysis with focus on transfer learning. Clinical Imaging.

Reviewer 2 Report

In this article, the authors analyze transfer Learning Strategy for Medical Object Detection on Brain MRI. The merits of this article include an in-depth analysis of previous research and a full presentation of the results.

However, there are points that can be improved:

It is necessary to describe in detail the software tools, and libraries used to conduct research so that readers can reproduce the research.

Author Response

Thank you very much for your time and effort to review our article.

Comment 1: It is necessary to describe in detail the software tools, and libraries used to conduct research so that readers can reproduce the research.

Response 1: Thank you for your valuable feedback. The relevant part has been updated.

All experiments were implemented in PyTorch.

Reviewer 3 Report

Congratulations to the authors for the results reported in the manuscript.

The research was well set up and the limits are unfortunately those highlighted by the authors themselves in the final chapter.

In particular having used only the BraTS dataset as a targeted TL source limits the final value of the system.

Author Response

Comment 1: The research was well set up and the limits are unfortunately those highlighted by the authors themselves in the final chapter. In particular having used only the BraTS dataset as a targeted TL source limits the final value of the system.

Response 1: Thank you for your valuable feedback. 

Future works will aim to minimize or eliminate any existing limitations as much as possible.

Reviewer 4 Report

The authors researched how transfer learning (TL) impacts medical object detection. Specifically, they investigated the effect of various weight initialization strategies (metrics) on the object detection task in Brain MRI with deep learning neural network (R-CNN). The proposed model was trained with five different weight initialization datasets (called strategies): Random, MS COCO, BraTS, MS COCO + BraTS, and MS COCO + BraTS + Fine Tune, using Gazi Brains 2020 dataset. The "success" of the pre-trained models in locating and segmenting the tumor region as an object recognition task on brain MRIs was compared, analyzed, and reported.

This is an interesting research report. The manuscript is well-structured, the research topic is actual, and appropriate for the Journal.

Before the publication, I have the following concerns/comments for the authors to address:

  1. I suggest the author's correct grammar (for example, L132, instead of "was", it should be "were") and improve the language. It would improve the article.
  2. L76-L93 should be deleted. There is no need to present results in the place where you define the aims. 
  3. Please, clearly state the study's aims (L69 - L71).
  4. Please define the acronym IoU before the first usage. 
  5. Chapter 4 Discussion is unclear. Please, rewrite it. It would add to the readability of the paper.

Author Response

We are grateful for your insightful comments and recommendations.

Comment 1: I suggest the author's correct grammar (for example, L132, instead of "was", it should be "were") and improve the language. It would improve the article.

Response 1: The relevant part has been updated.

“In this study, both TL strategies weight initialization and fine tuning mechanism were used as the learning transfer method, since TL effect was measured.”

Comment 2: L76-L93 should be deleted. There is no need to present results in the place where you define the aims. 

Response 2: The relevant part has been deleted.

Comment 3: Please, clearly state the study's aims (L69 - L71).

Response 3: It has been added in line 84-88.

Comment 4: Please define the acronym IoU before the first usage. 

Response 4: The relevant part has been updated. 

“The X-axis represents the epoch, and the Y-axis represents the Intersection over Union (IoU) value corresponding to the specified metric.”

Comment 5: Chapter 4 Discussion is unclear. Please, rewrite it. It would add to the readability of the paper.

Response 5: The discussion section has been rewritten by combining it with the conclusion.

Reviewer 5 Report

Dear Authors,

this is a well-written manuscript, presenting the evaluation of Transfer Learning strategies for the detection of medical objects on brain MRI. I have no questions or queries pertaining to your manuscript.

Best Regards

Author Response

Comment 1: This is a well-written manuscript, presenting the evaluation of Transfer Learning strategies for the detection of medical objects on brain MRI. I have no questions or queries pertaining to your manuscript.

Response 1: Thank you for your valuable feedback.

Round 2

Reviewer 1 Report

 no further comments

Author Response

Thank you for your valuable reviewing process.

Reviewer 2 Report

The author took into account the comments.

The results of the experiments are presented in an accessible form.

For an even better understanding, it is recommended to describe the used libraries and their versions in more detail.

Author Response

Comment:

The author took into account the comments.

The results of the experiments are presented in an accessible form.

For an even better understanding, it is recommended to describe the used libraries and their versions in more detail.

Response:

Thank you for your comments that make the Manuscript more impressive. We've added the relevant library information to the first paragraph of the Results Section.